# Buongiorno's Nanofluid Model over a Curved Exponentially Stretching Surface

**Adel Alblawi [1], Muhammad Yousaf Malik [2], Sohail Nadeem [3] and Nadeem Abbas [3],***

[1]  Mechanical Engineering Department, College of Engineering, Shaqra University, Dawadmi, P.O. 11911, Ar Riyadh 11564, Saudi Arabia; aalblawi@su.edu.sa
[2]  Department of Mathematics, College of Sciences, PO Box 9004, King Khalid University, Abha 61413, Saudi Arabia; drmymalik@hotmail.com
[3]  Department of Mathematics, Quaid-I-Azam University 45320, Islamabad 44000, Pakistan; sohail@qau.edu.pk
*   Correspondence: nabbas@math.qau.edu.pk

**Abstract:** We considered the steady flow of Buongiorno's model over a permeable exponentially stretching channel. The mathematical model was constructed with the assumptions on curved channels. After applying the boundary layer approximation on the Navier–Stocks equation, we produced nonlinear partial differential equations. These equations were converted into a system of non-dimensional ordinary differential equations through an appropriate similarity transformation. The dimensionless forms of the coupled ordinary differential equations were elucidated numerically through boundary value problem fourth order method. This method gains fast convergence as compared to other method such as the shooting method and the Numerical Solution of Differential Equations Mathematica method. The influence of the governing parameters which are involved in ordinary differential equations are highlighted through graphs while $Re_s^{1/2}C_f$, $Re_s^{1/2}N_{u_s}$, and $Re_s^{-1/2}Sh_s$ are highlighted through the tables. Our interest of study was to analyze the heat transfer rate of nanofluids. Surprisingly, for momentum boundary layer thickness, thermal boundary layer thickness and solutal boundary layer thickness became larger when $\lambda > 0$, as compared to the case when $\lambda < 0$.

**Keywords:** Buongiorno's model; thermal slip effects; exponential stretching; numerical technique; curved channel

## 1. Introduction

Analysis of stretching surfaces play a key role in the field of engineering and industrial due to its practical applications. The boundary layer, defined as the viscosity effects, are important nearby the surface. The boundary layer concept was highlighted by Ludwig Prandtl in 12 August 1904 at the 3rd International Conference in Germany. He considered two regions for the fluid—where viscosity effects are maximized inside the boundary, while viscosity effects are negligible outside the boundary. At the continuous moving solid surface, a two-dimensional flow of a Newtonian fluid was highlighted by Sakiadis [1]. He was the first one who discussed the boundary layer flow of Newtonian fluid. His study developed great interest and attracted researchers to analyze boundary layer flow. Crane [2] extended the idea of Sakiadis [1]. Crane [2] deliberated over the exact solution of quiescent fluid over a linearly stretching sheet. The stagnation point within flow over a stretching surface has been studied by Chiam [3]. Where he analyzed particular flow geometries. Lin et al. [4] considered two-dimensional boundary layer flows with time dependent heat flux. Their study focused on a large boundary layer and small Prandtl numbers using hypergeometric functions. Chen [5] studied the mixed convection flow over a heated stretching surface. He emphasized the behavior of various physical parameters and

designed figures (graphs) for skin friction and the local Nusselt number. Mahapatra and Gupta [6] found the solution numerically over a stretching surface under the stagnation point. An influence of buoyancy on boundary layer flow of a continuous stretching surface was highlighted by Ali [7]. Wang [8] investigated the flow toward a shrinking sheet under the stagnation point. He found that a solution does not exist for high enough shrinking rates and when it does exist, it may not be unique in two dimensions. Analysis of the viscous flow toward a shrinking sheet with suction and slip effects was explored by Wang [9]. A power law fluid model over a stretching sheet under the stagnation point was studied by Mahapatra et al. [10]. Mahapatra et al. [10] used numerical and analytical techniques and presented a comparison of both solutions. Analysis of the unsteady flow of fluid over a stretching cylinder was investigated by Tie-Gang et al. [11]. Wang [12] discussed the natural convection over a vertical cylinder. Salahuddin et al. [13] discussed the flow of nanofluids on a stretching cylinder near the stagnation region. Several researchers have explored the flow in a curved channel [14–16]; however, none have considered exponentially curved channel geometry.

Nanofluids have notable thermal characteristics and hence are useful for heat transfer applications. A nanofluid is a combination of a homogeneous fluid (base) and nanomaterials. There are numerous applications such as heat exchangers, automotive cooling applications, and technology plants. The nanofluid concept was presented by Choi [17]. He projected an innovative class of heat transfer fluid where an imaginative new class of heat transfer liquids can be built by suspending metallic nanoparticles in conventional heat transfer fluids. The subsequent nanofluids show high thermal conductivities in contrast with presently utilized heat transfer fluids and they epitomize the results in the improvement of heat transfer fluid. Jang and Choi [18] proposed that Brownian Motion can improve thermal conductivity. They showed that the Brownian movement of nanoparticles at the nanoscale and molecular levels is a key system overseeing the thermal conduct of nanoparticle fluid suspensions. Convective transport in nanofluids was pioneered by Buongiorno [19]. According to him, the nanofluids are designed colloids made of a base fluid and nanoparticles (1–100 nm). Nanofluids have a greater thermal physical phenomenon than that of simple fluid. Precisely, the heat transfer coefficient expansion seems to go beyond the important thermal conductivity effect, and cannot be foreseen by conventional pure fluid relationships, for example, Dittus–Boelter's. He deliberated, therefore, seven slip mechanisms: Inertia, Brownian diffusion, thermo phoresis, Magnus impact, diffusion phoresis, fluid drainage, and gravity, and asserted while Brownian diffusion are significant slip systems in nanofluids. Oztop and Nada [20] analyzed the natural convection of nanofluid due to buoyancy forces in a partially heated rectangle. Numerical studies of laminar nanofluid flow by two isothermally heated parallel plates were performed by Santra et al. [21]. Khan and Pop [22] pioneered the nanofluid flow over a stretching sheet. Nadeem et al. [23] introduced the peristaltic flow of nanofluid over curved channels. Several researchers have highlighted nanofluid flow on curved channels influenced by several parameters [24–27].

The no-slip boundary condition is a foundational aspect within the fluid dynamics theory of Navier–Stokes. The no-slip condition is insufficient for most non-Newtonian liquids and nanofluids, as some polymer melt often demonstrates microscopic wall slip and is generally regulated by a nonlinear and monotonous relationship between slip velocity and traction. The fluids showing limit slip have applications in innovative issues, for example, the cleaning of artificial heart valves and inside cavities. Slip condition on the boundary layer flow was introduced by Andersson [28]. He pioneered the closed form results of the Navier–Stokes equations over stretching sheets with MHD flow. Using the Andersson [28] idea, Wang [29] pioneered the closed form solution of the Navier–Stokes equations over stretching sheets with a slip condition. Moreover, stagnation flow with a slip condition was initiated by the Wong [30]. Fang et al. [31] found the exact solution of viscous flow over stretching sheets having influence of slip and MHD. Several researchers are worked out the slip condition with different aspects [32–35].

In this study we analyzed the impact of different physical significant parameters of hybrid nanofluid over an exponentially stretching curved surface. Our mathematical model of flow consisted

of a system of partial differential equations, which we then transformed into an ordinary differential equation using similarity transformations. These equations were solved by the numerical bvp4c method. We also highlighted the influence of the curvature parameter, thermal slip parameter, suction/injection parameter, solid nanoparticle, and stretching parameter on the hybrid nanofluid. The expression of the skin friction and Nusselt numbers were used to understand the flow properties. This analysis provides innovative insight for applications and complements the existing literature.

## 2. Mathematical Formulations

We considered the Buongiorno fluid model over a permeable stretching channel. Arc length along the flow direction is *s* and normal to tangent vector is *r* Figure 1a. The geometry of the stretching is revealed in Figure 1b.

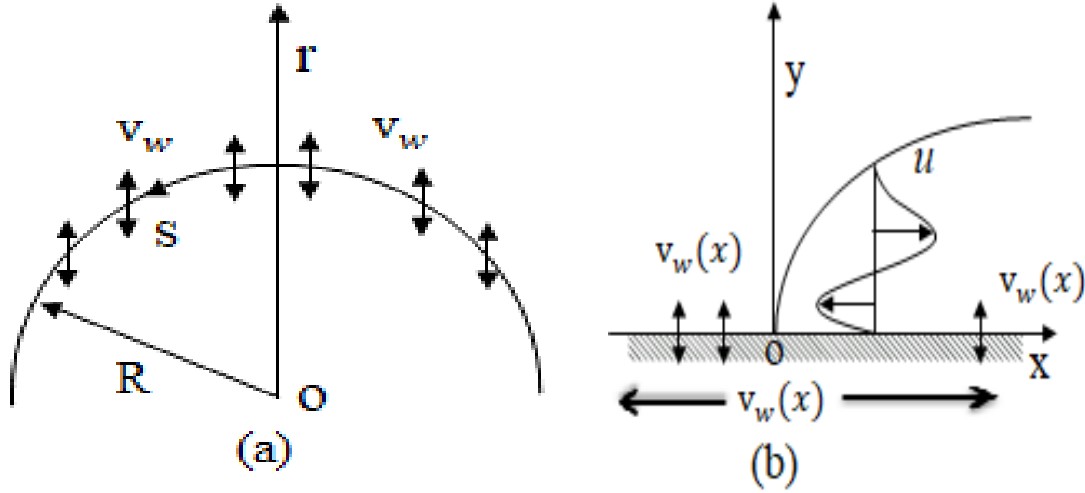

**Figure 1.** Physical model and coordinates system: (**a**) curvature surface; (**b**) stretching surface.

The surface is stretched where $v_w = ce^{\frac{s}{a}}$ in the − direction and c is constant while, $v_w$ is suction\injection parameter due to porous surface which represents two cases if $v_w < 0$ and $v_w > 0$ correspond to injection and suction, respectively. But we considered in our analysis $v_w > 0$, such as suction. Our mathematical model includes the following, having used the above assumptions, boundary layer approximations, and governing equations for the flow,

$$\frac{1}{r+R}\frac{\partial}{\partial r}((r+R)v) + \frac{R}{r+R}\frac{\partial u}{\partial s} = 0, \tag{1}$$

$$\frac{u^2}{r+R} = \frac{1}{\rho_f}\frac{\partial p}{\partial r}, \tag{2}$$

$$\left(v\frac{\partial u}{\partial r} + \frac{Ru}{r+R}\frac{\partial u}{\partial s} + \frac{vu}{r+R}\right) + \frac{1}{\rho_f}\frac{1}{r+R}\frac{\partial p}{\partial r} = \frac{\mu_f}{\rho_f}\left(\frac{\partial^2 u}{\partial^2 r} + \frac{1}{r+R}\frac{\partial u}{\partial r} - \frac{u}{(r+R)^2}\right), \tag{3}$$

$$\left(v\frac{\partial T}{\partial r} + \frac{Ru}{r+R}\frac{\partial T}{\partial s}\right) = \alpha_f\left(\frac{\partial^2 T}{\partial^2 r} + \frac{1}{r+R}\frac{\partial T}{\partial r}\right) + \left(\frac{\rho c_p}{\rho c_f}\right)\left(D_B\left(\frac{\partial C}{\partial r}\right)\left(\frac{\partial T}{\partial r}\right) + \left(\frac{D_T}{T_\infty}\right)\left(\frac{\partial T}{\partial r}\right)^2\right), \tag{4}$$

$$\left(v\frac{\partial C}{\partial r} + \frac{Ru}{r+R}\frac{\partial C}{\partial s}\right) = D_B\left(\frac{\partial^2 C}{\partial^2 r} + \frac{1}{r+R}\frac{\partial C}{\partial r}\right) + \left(\frac{D_T}{T_\infty}\right)\left(\frac{\partial^2 T}{\partial^2 r} + \frac{1}{r+R}\frac{\partial T}{\partial r}\right), \tag{5}$$

with respected boundary conditions

$$u = ce^{\frac{s}{a}}, v = v_w, T = T_w + \lambda_1\left(\frac{\partial T}{\partial r}\right), D_B\left(\frac{\partial C}{\partial r}\right) + \left(\frac{D_T}{T_\infty}\right)\left(\frac{\partial T}{\partial r}\right) = 0, \text{ at } r \to 0,$$
$$u \to u_w, \text{ at } r \to 0, T \to T_\infty, C \to C_\infty, \text{ at } r \to \infty. \tag{6}$$

The following similarity transformations have been highlighted as below

$$
T = T_w + c\theta(\zeta),\ \zeta = \sqrt{\tfrac{a}{\nu_f}}r,\ u = a\left(e^{\frac{s}{a}}\right)F'(\zeta),\ v = -\tfrac{R}{r+R}\sqrt{\tfrac{\nu_f}{a}}\left(e^{\frac{s}{a}}\right)F(\zeta),
$$
$$
C = C_w + c\phi(\zeta),\ P = \rho a^2\left(e^{\frac{2s}{a}}\right)p(\zeta).
$$
(7)

The dimensionless suitable similarity transformations are applied on the Equations (1)–(6). The density, viscosity, thermal diffusivity of the fluid is noted as $\rho_f$, $\mu_f$ and $\alpha_f$ respectively; $p$ is a pressure, R is the radius of curvature, $T_\infty$ is the ambient temperature and $T_w$ is the wall temperature. The partial differential equations are altered into ordinary differential equations by applying the similarity transformation. The reduced system is

$$
P' = \frac{F'^2}{\zeta + K_0}
$$
(8)

$$
\frac{2K}{\zeta + K_0}P = \left[F''' + \frac{F''}{\zeta + K_0} - \frac{F'}{(\zeta + K_0)^2}\right] - \frac{R_0 K_0}{\zeta + K_0}F'^2 + \frac{R_0 K_0}{\zeta + K_0}F'F'' + \frac{R_0 K_0}{(\zeta + K_0)^2}FF',
$$
(9)

$$
\frac{1}{Pr}\left(\theta'' + \frac{1}{\zeta + K_0}\theta'\right) + \frac{K_0 R_0}{\zeta + K_0}F\theta' - \frac{K_0 R_0}{\zeta + K_0}\theta + R_0(N_B\theta'\phi' + N_T\theta'\theta') = 0,
$$
(10)

$$
\left(\phi'' + \frac{1}{\zeta + K_0}\phi'\right) + \frac{K_0 R_0}{\zeta + K_0}F\phi' - \frac{K_0 R_0}{\zeta + K_0}\phi + \frac{N_B}{N_T}\left(\theta'' + \frac{1}{\zeta + K_0}\theta'\right) = 0,
$$
(11)

Eliminating the pressure term, solving Equations (8) and (9)

$$
\left(F'''' + \frac{2}{\zeta + K_0}F''' - \frac{1}{(\zeta + K_0)^2}F'' + \frac{1}{(\zeta + K_0)^3}F'\right) - \frac{R_0 K_0}{\zeta + K_0}\left(F''F' - FF'''\right)
$$
$$
- \frac{R_0 K_0}{(\zeta + K_0)^2}\left(F'^2 - FF''\right) - \frac{R_0 K_0}{(\zeta + K_0)^3}FF' = 0,
$$
(12)

$$
\frac{1}{Pr}\left(\theta'' + \frac{1}{\zeta + K_0}\theta'\right) + \frac{K_0 R_0}{\zeta + K_0}F\theta' - \frac{K_0 R_0}{\zeta + K_0}\theta + R_0 N_B\theta'\phi' + R_0 N_T\theta'\theta' = 0,
$$
(13)

$$
\left(\phi'' + \frac{1}{\zeta + K_0}\phi'\right) + \frac{K_0 R_0}{\zeta + K_0}F\phi' - \frac{K_0 R_0}{\zeta + K_{0,}}\phi + \frac{N_B}{N_T}\left(\theta'' + \frac{1}{\zeta + K_0}\theta'\right) = 0,
$$
(14)

Dimensionless boundary conditions take the form

$$
F(0) = \gamma,\ F'(0) = \lambda + S(F''(0) - F'(0)/K_0),\ F'(\infty) = 1,
$$
$$
F''(\infty) = 0,\ N_B\phi'(0) + N_T\theta'(0) = 0,
$$
$$
M\theta'(0) + 1 = \theta(0),\ \theta(\infty) = 0,\ \phi(\infty) = 0.
$$
(15)

where in the case of, if $\gamma < 0$ is the suction and $\gamma > 0$ is the injection, $\beta$ represents the stretching parameter, $M$ is the thermal slip parameter, $k_f$ be the thermal conductivity of fluid, the curvature parameter is $K_0$, the Brownian motion parameter $N_B$, the thermophoresis parameter $N_T$ and $R_0$ is the dimensionless parameter. The physical features of the interest are the Nusselt number and skin friction coefficients along the $s$ direction, which are highlighted below

$$
C_f = \frac{\tau_{rs}}{\rho_f u_w^2},\quad N_s = \frac{sq_w}{k_f(T_w - T_\infty)},\quad Sh_s = \frac{sh_m}{D_B(C_w - C_\infty)},
$$
(16)

where $\tau_{rs}$ and $q_w$ is wall shear stress and heat flux, respectively, at the wall in $s$-direction. The expression of the wall shear stress and heat flux are to be defined as

$$
\tau_{rs} = \left(\frac{\partial u}{\partial r} - \frac{u}{r + R}\right)_{r=0},\quad q_w = -\left(\frac{\partial T}{\partial r}\right)_{r=0},\quad h_m = -\left(\frac{\partial C}{\partial r}\right)_{r=0},
$$
(17)

Using Equation (18) in Equation (17), we get the dimensionless form as follows

$$Re_s^{-1/2}C_f = \left(F''(0) - \frac{F'(0)}{K}\right) \tag{18}$$

$$Re_s^{-1/2}N_{u_s} = -\Theta'(0) \tag{19}$$

$$Re_s^{-1/2}Sh_s = -\phi'(0) \tag{20}$$

where $Re_s = \left(\dfrac{ae^{\frac{2s}{a}}}{\nu_f}\right)$ is the local Reynolds number.

## 3. Results and Discussion

The steady flow of mixed convection over a curved channel is considered in this study. We developed a mathematical model that we solved numerically using the bvp4c method. The physical parameters which are involved in the fluid flow behavior have been highlighted through graphs and tables. The impact of the Brownian motion parameter $N_B$, the thermophoresis parameter $N_T$, non-dimensional parameter $R_0$, micropolar parameter $K_0$, thermal slip parameter $M$, suction parameter $\gamma$, stretching or shrinking parameter $\lambda$, and velocity slip $S$ on velocity profile, concentration profile and temperature profile are revealed in Figures 2–17.

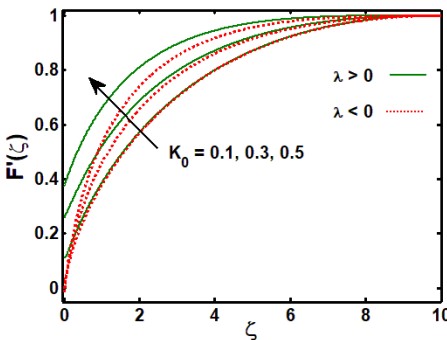

**Figure 2.** Effects of the $R_0$ and $K_0$ on the $F'(\zeta)$.

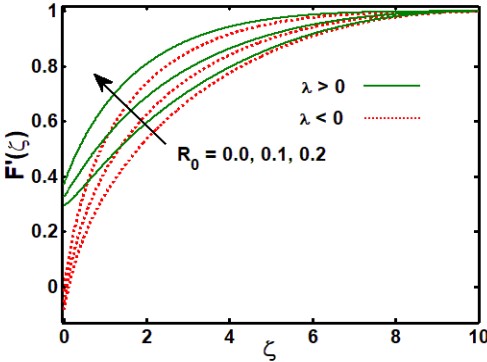

**Figure 3.** Effects of the $R_0$ on the $F'(\zeta)$.

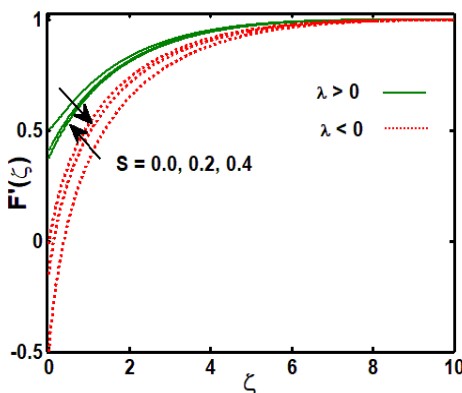

**Figure 4.** Effects of the $S$ on the $F'(\zeta)$.

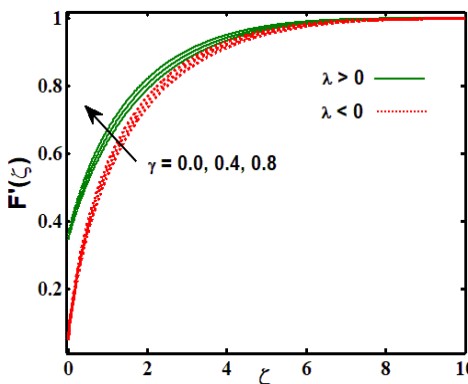

**Figure 5.** Effects of the $\gamma$ on the $F'(\zeta)$.

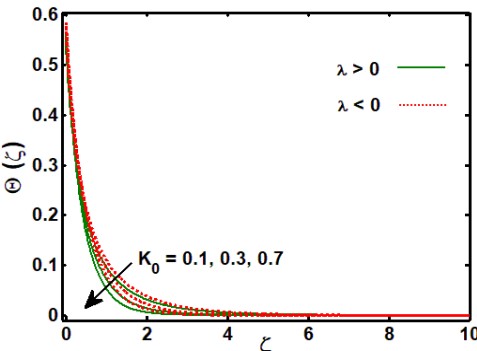

**Figure 6.** Effects of $K_0$ on the $\Theta(\zeta)$.

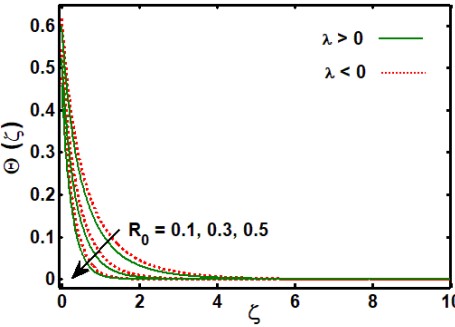

**Figure 7.** Effects of $R_0$ on the $\Theta(\zeta)$.

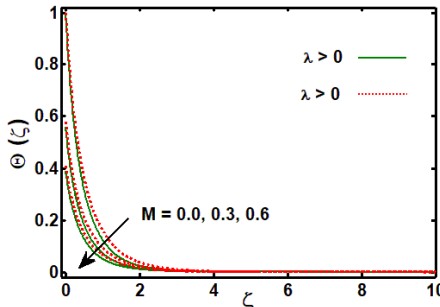

**Figure 8.** Effects of the M on the $\Theta(\zeta)$.

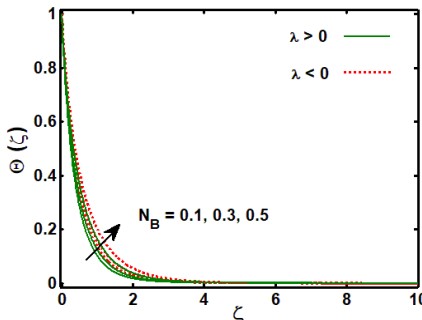

**Figure 9.** Effects of $N_B$ on the $\Theta(\zeta)$.

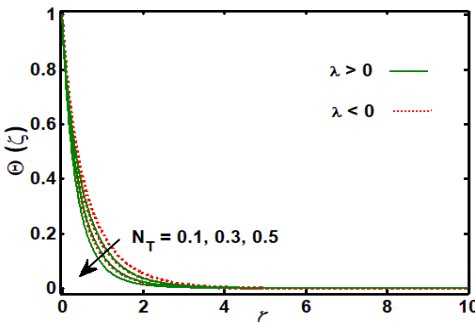

**Figure 10.** Effects of $N_T$ on the $\Theta(\zeta)$.

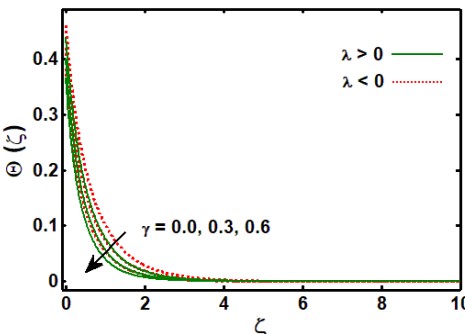

**Figure 11.** Effects of $\gamma$ on the $\Theta(\zeta)$.

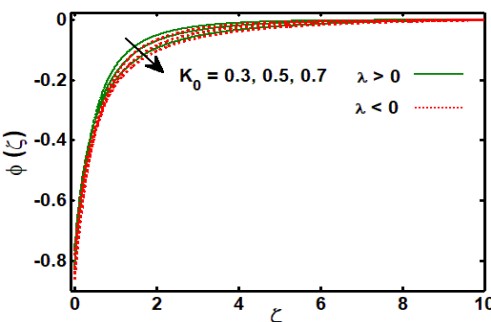

**Figure 12.** Effects of $K_0$ on the $\phi(\zeta)$.

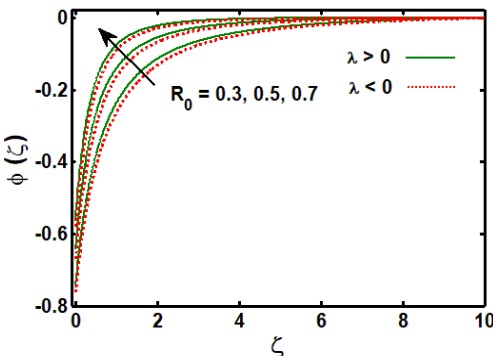

**Figure 13.** Effects of $R_0$ on the $\phi(\zeta)$.

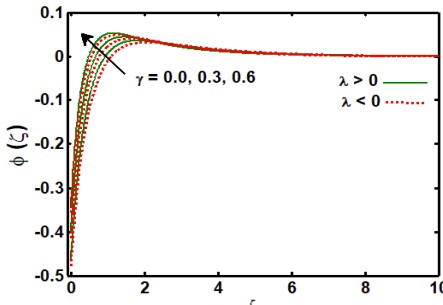

**Figure 14.** Effects of $\gamma$ on the $\phi(\zeta)$.

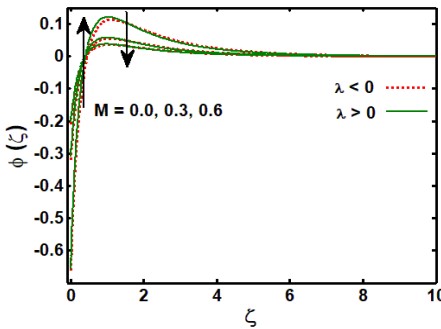

**Figure 15.** Effects of $M$ on the $\phi(\zeta)$.

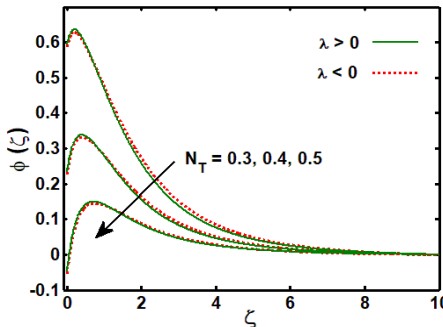

**Figure 16.** Effects of $N_T$ on the $\phi(\zeta)$.

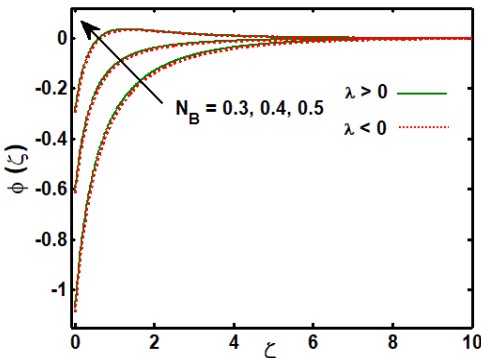

**Figure 17.** Effects of $N_B$ on the $\phi(\zeta)$.

### 3.1. Influence of Physical Parameters on Velocity Profile

Here we have shown the effects of the non-dimensional parameter $R_0$, micropolar parameter $K_0$, suction $\gamma$, stretching/shrinking parameter $\lambda$, the Brownian motion parameter $N_B$, the thermophoresis parameter $N_T$, and velocity slip $S$ on velocity profile $F'(\zeta)$. Figure 2 shows the impacts of physical parameter $K_0$ on the velocity profile. It illustrates that the momentum boundary layer thickness improves when the value of $K_0$ increases. In the both cases of stretching or shrinking parameter $\lambda$, the velocity profile increases but the stretching parameter gains higher momentum boundary layer thickness when compared to shrinking parameter. Figure 3 displays the impact of $R_0$ on the velocity profile. The momentum boundary layer thickness increases as $R_0$ increases. It is highlighted that achieves of the momentum boundary layer thickness is more sensitive to the stretching parameter than the shrinking parameter. The impact of velocity slip on the velocity profile is highlighted in Figure 4. It is seen that velocity profile is increased with an increase in the velocity slip parameter in the case of $\lambda > 0$ and declines the velocity profile when the velocity slip parameter is increased in case of $\lambda < 0$. Figure 5 reveals the impact of the suction parameter on the velocity profile. It is observed that velocity profile increases for the higher value of the suction parameter. In the both cases of the stretching or shrinking parameter, the momentum boundary layer thickness increases for the higher values of the suction parameter.

### 3.2. Effects of Physical Parameters on Temperature Profile

Figures 6–11 reveal the effects of physical parameters on the temperature profile. The influences of $K_0$ and $R_0$ on the temperature profile is highlighted in Figures 6 and 7. It is noted that thermal boundary layer thickness declines when the values of $K_0$ and $R_0$ increase. But in case of a stretching surface, the values of the thermal boundary layer thickness increases when compared to a shrinking surface. Figures 8 and 9 highlight the impact of M and $N_B$ on the temperature profile. The thermal boundary layer thickness declines for larger values of M, but in case of stretching surface, the values of the thermal boundary layer thickness achieves fastly when compared to a shrinking surface, while the

opposite is observed for the effect of $N_B$ on the the temperature profile, shown in Figure 9. The impact of $\gamma$ and $N_T$ on the temperature profile is highlighted in Figures 10 and 11. The thermal boundary layer thickness declines for the larger values of $\gamma$ and $N_T$. In case of a stretching surface, the values of the thermal boundary layer thickness increases when compared to a shrinking surface, as highlighted in in Figures 10 and 11.

### 3.3. Effects of Physical Parameters on Concentration Profile

Figures 12–17 reveal the influence of physical parameters on concentration profile which illustrates interesting results. The influence of the $K_0$ and $R_0$ on the concentration profile is highlighted in Figures 12 and 13. The values of $K_0$ increased while the concentration profile reduced, as highlighted in Figure 12. In the case of a stretching surface, the values of solutal layer thickness increased when it compared with a shrinking surface. Figure 13 highlights the influence of $R_0$ on the concentration profile. It is noted that the concentration profile increased the solutal layer thickness when the values of $R_0$ increased in case of a stretching surface, the values of solutal layer thickness increased when it compared with shrinking surface. Figures 14 and 15 reveal the influence of $\gamma$ and $M$ on the concentration profile. As $\gamma$ increased, the concentration profile increased. It is very interesting that the solutal layer thickness was raised as the values of $\gamma$ while in case of a stretching surface, the values of solutal layer thickness achieves fastly when compared with a shrinking surface, which is shown in Figure 14. Figure 15 highlights the impact of $M$ on the concentration profile. It is prominent that the values of $M$ increased while the solutal layer thickness increased and after the intersection point, the opposite behavior was seen in the case of the stretching surface; the values of solutal layer thickness increased when it compared with the shrinking surface. The impact of $N_B$ and $N_T$ on the concentration profile is highlighted in Figures 16 and 17. The concentration profile reduced as the values of $N_T$ rose. The solutal layer thickness increased for the higher values of $N_T$ but after the point of the intersection, the solutal layer thickness changed the behavior for $N_T$ as reduce for the larger values of $N_T$. Figure 17 reveals the influence of $N_T$ on the concentration profile. It is seen that the concentration profile increased for greater values of $N_T$ while in case of stretching surface, the values of solutal layer thickness increased when compared to the shrinking surface.

### 3.4. Numerical Results

Table 1 displays the impact of physical parameters namely, non-dimensional parameter $R_0$, curvature parameter $K_0$, thermal slip parameter $M$, suction parameter $\gamma$, stretching or shrinking parameter $\lambda$, the Brownian motion parameter $N_B$, the thermophoresis parameter $N_T$, and velocity slip $S$, as highlighted on the $Re_s^{1/2}C_f$, $Re_s^{1/2}Nu_s$, and $Re_s^{-1/2}Sh_s$. The values of $Re_s^{1/2}C_f$ rises for the higher values of $R_0$ in case of $\lambda < 0$ but reduces in case of $\lambda > 0$. The values of $Re_s^{1/2}Nu_s$ and $Re_s^{-1/2}Sh_s$ enhances for larger values of $R_0$ in both cases of $\lambda > 0$ and $\lambda < 0$. The values of $K_0$ rises with the declining of $Re_s^{1/2}C_f$, $Re_s^{1/2}Nu_s$, and $Re_s^{-1/2}Sh_s$ in case of $\lambda > 0$ while reduces the values of $Re_s^{1/2}C_f$ and increases the values of $Re_s^{1/2}Nu_s$ and $Re_s^{-1/2}Sh_s$ for large values of physical parameter $K_0$ in case of $\lambda < 0$. It is noted that the values of $Re_s^{1/2}C_f$ decline in case of $\lambda > 0$ while increase the values of $Re_s^{1/2}C_f$ in case of $\lambda < 0$ for the larger values of $\lambda$. Also highlighted are the effects of $\lambda$ on the $Re_s^{1/2}Nu_s$ and $Re_s^{-1/2}Sh_s$. The values of $Re_s^{1/2}Nu_s$ and $Re_s^{-1/2}Sh_s$ rises with the rising values of $\lambda$ in both cases of $\lambda > 0$ and $\lambda < 0$. The numerical values f $Re_s^{1/2}Nu_s$ and $Re_s^{-1/2}Sh_s$ decline for the greater values of $M$ in both cases of $\lambda > 0$ and $\lambda < 0$. The numerical values of $Re_s^{1/2}Nu_s$ and $Re_s^{-1/2}Sh_s$ decline for the greater values of $N_B$ in both cases of $\lambda > 0$ and $\lambda < 0$. The numerical values f $Re_s^{1/2}Nu_s$ and $Re_s^{-1/2}Sh_s$ increase for the greater values of $N_B$ in both cases of $\lambda > 0$ and $\lambda < 0$. The numeral values of $Re_s^{1/2}C_f$, $Re_s^{1/2}Nu_s$ and $Re_s^{-1/2}Sh_s$ decline for the higher values of $S$ in the case of $\lambda > 0$. It is also interesting that the values of $Re_s^{1/2}C_f$ decline and the values of $Re_s^{1/2}Nu_s$ and $Re_s^{-1/2}Sh_s$ increase for higher values of $S$. Table 2 shows the comparison with different method to find the solution. It is observed that the BVP4C method finds to best agreement as compared to other methods.

**Table 1.** Numerical results of physical parameters for $Re_s^{1/2}C_f$, $Re_s^{1/2}Nu_s$, and $Re_s^{-1/2}Sh_s$.

| | | | | | | | For $\lambda > 0$ | | | For $\lambda < 0$ | | |
|---|---|---|---|---|---|---|---|---|---|---|---|---|
| $R_0$ | $K_0$ | $\gamma$ | $M$ | $N_B$ | $N_T$ | $S$ | $Re_s^{1/2}C_f$ | $Re_s^{1/2}Nu_s$ | $Re_s^{-1/2}Sh_s$ | $Re_s^{1/2}C_f$ | $Re_s^{1/2}Nu_s$ | $Re_s^{-1/2}Sh_s$ |
| 0.1 | 0.3 | 0.4 | 0.4 | 0.5 | 0.5 | 0.4 | −0.6559 | 1.0014 | −1.0014 | 1.1634 | 0.9934 | −0.9934 |
| 0.2 | | | | | | | −0.6056 | 1.1639 | −1.1639 | 1.2110 | 1.1527 | −1.1527 |
| 0.3 | | | | | | | −0.5417 | 1.2764 | −1.2764 | 1.2710 | 1.2635 | −1.2635 |
| 0.4 | | | | | | | −0.4584 | 1.3642 | −1.3642 | 1.3488 | 1.3502 | −1.3502 |
| 0.3 | 0.1 | | | | | | −0.9736 | 1.4909 | −1.4909 | 1.2485 | 1.4890 | −1.4890 |
| | 0.2 | | | | | | −0.7471 | 1.3385 | −1.3385 | 1.2516 | 1.3315 | −1.3315 |
| | 0.3 | | | | | | −0.5417 | 1.2764 | −1.2764 | 1.2710 | 1.2635 | −1.2635 |
| | 0.4 | | | | | | −0.3472 | 1.2454 | −1.2454 | 1.3062 | 1.2264 | −1.2264 |
| | 0.3 | 0.0 | | | | | −0.5593 | 1.1712 | −1.1712 | 1.2537 | 1.1572 | −1.1572 |
| | | 0.2 | | | | | −0.5507 | 1.2244 | −1.2244 | 1.2622 | 1.2109 | −1.2109 |
| | | 0.4 | | | | | −0.5417 | 1.2764 | −1.2764 | 1.2710 | 1.2635 | −1.2635 |
| | | 0.6 | | | | | −0.5324 | 1.3269 | −1.3269 | 1.2802 | 1.3146 | −1.3146 |
| | | 0.4 | 0.0 | | | | −0.5417 | 2.5835 | −2.5835 | 1.2710 | 2.5312 | −2.5312 |
| | | | 0.2 | | | | −0.5417 | 1.7104 | −1.7104 | 1.2710 | 1.6873 | −1.6873 |
| | | | 0.4 | | | | −0.5417 | 1.2764 | −1.2764 | 1.2710 | 1.2635 | −1.2635 |
| | | | 0.6 | | | | −0.5417 | 1.0175 | −1.0175 | 1.2710 | 1.0093 | −1.0093 |
| | | | 0.4 | 0.1 | | | −0.5417 | 1.3110 | −6.5549 | 1.2710 | 1.2980 | −6.4898 |
| | | | | 0.3 | | | −0.5417 | 1.2991 | −2.1652 | 1.2710 | 1.2861 | −2.1435 |
| | | | | 0.5 | | | −0.5417 | 1.2764 | −1.2764 | 1.2710 | 1.2635 | −1.2635 |
| | | | | 0.7 | | | −0.5417 | 1.2449 | −0.8892 | 1.2710 | 1.2322 | −0.8801 |
| | | | | 0.5 | 0.1 | | −0.5417 | 1.1503 | −0.2301 | 1.2710 | 1.1385 | −0.2277 |
| | | | | | 0.3 | | −0.5417 | 1.2438 | −0.7463 | 1.2710 | 1.2311 | −0.7387 |
| | | | | | 0.5 | | −0.5417 | 1.2764 | −1.2764 | 1.2710 | 1.2635 | −1.2635 |
| | | | | | 0.7 | | −0.5417 | 1.2985 | −1.8179 | 1.2710 | 1.2855 | −1.7997 |
| | | | | | 0.5 | 0.0 | −1.9708 | 1.2864 | −1.2864 | 4.6233 | 1.2388 | −1.2388 |
| | | | | | | 0.2 | −0.8498 | 1.2786 | −1.2786 | 1.9939 | 1.2582 | −1.2582 |
| | | | | | | 0.4 | −0.5417 | 1.2764 | −1.2764 | 1.2710 | 1.2635 | −1.2635 |
| | | | | | | 0.6 | −0.3976 | 1.2754 | −1.2754 | 0.9328 | 1.2659 | −1.2659 |

**Table 2.** Comparison equeaBVP4C with shooting method and ND solve method.

| Parameter | | | BVP4C Method | | Shooting Method | | ND Solve Method | |
|---|---|---|---|---|---|---|---|---|
| $M$ | $N_B$ | $N_T$ | $Re_s^{1/2}Nu_s$ | $Re_s^{-1/2}Sh_s$ | $Re_s^{1/2}Nu_s$ | $Re_s^{-1/2}Sh_s$ | $Re_s^{1/2}Nu_s$ | $Re_s^{-1/2}Sh_s$ |
| 0.0 | 0.5 | 0.5 | 2.5835 | −2.5835 | 2.5835 | −2.5765 | 1.9865 | −2.4689 |
| 0.2 | | | 1.7104 | −1.7104 | 1.6812 | −1.7011 | 1.5864 | −1.6841 |
| 0.4 | | | 1.2764 | −1.2764 | 1.2584 | −1.2698 | 1.1981 | −1.2963 |
| 0.6 | | | 1.0175 | −1.0175 | 1.0115 | −1.0109 | 0.9987 | −1.0115 |
| 0.4 | 0.1 | | 1.3110 | −6.5549 | 1.2986 | −6.4986 | 1.0982 | −6.3124 |
| | 0.3 | | 1.2991 | −2.1652 | 1.2869 | −2.0985 | 1.1978 | −2.0258 |
| | 0.5 | | 1.2764 | −1.2764 | 1.2689 | −1.1989 | 1.2114 | −1.0638 |
| | 0.7 | | 1.2449 | −0.8892 | 1.2334 | −0.7995 | 1.1983 | −0.6987 |
| | 0.5 | 0.1 | 1.1503 | −0.2301 | 1.1369 | −0.1968 | 1.0953 | −0.1657 |
| | | 0.3 | 1.2438 | −0.7463 | 1.2589 | −0.7328 | 1.1896 | −0.6985 |
| | | 0.5 | 1.2764 | −1.2764 | 1.2765 | −1.2654 | 1.2546 | −1.1896 |
| | | 0.7 | 1.2985 | −1.8179 | 1.2893 | −1.7691 | 1.2789 | −1.7361 |

## 4. Conclusions

We considered the steady flow of Buongiorno's model over a permeable exponentially stretching channel. This is the first mathematical model to focus on an exponentially stretching, curved channel. The mathematical model was solved numerically through a BVP4C scheme. The influence of the governing parameters which were involve in ordinary differential equations were highlighted through graphs while $Re_s^{1/2}C_f$, $Re_s^{1/2}Nu_s$, and $Re_s^{-1/2}Sh_s$ were highlighted through the tables. The following interesting results were obtained given the assumptions: Momentum boundary layer thickness, thermal boundary layer thickness, and solutal boundary layer thickness grow larger when $\lambda > 0$ as compared to the case when $\lambda < 0$. Numerical values are compared with three methods, namely, the BVP4C method, the shooting technique, and the ND solve technique in Table 2. It is noted that the BVP4C is a much better technique when compared to the shooting technique and the ND solve method.

**Author Contributions:** N.A. and S.N. designed the model and write-up, M.Y.M. help in the mathematical model, A.A. helps in the revised manuscript.

**Funding:** This research received no external funding.

**Acknowledgments:** The authors would like to express their gratitude to King Khalid University, Abha 61413, Saudi Arabia for providing administrative and technical support.

**Conflicts of Interest:** The authors declare no conflict of interest.

## Nomenclature

Physical parameters

| | | | |
|---|---|---|---|
| Pr | Prandtl number | $T_w$ | Wall temperature |
| $\gamma$ | Suction/Injection | $T_\infty$ | Ambient temperature |
| $M$ | Thermal slip | $\nu_f$ | Fluid kinematic Viscosity |
| $K_0$ | Curvature parameter | $\kappa_f$ | Thermal conductivity of fluid |
| $R_0$ | Stretching parameter | $\lambda$ | Stretching or shrinking parameter |
| $\Theta$ | Temperature profile | $S$ | Velocity slip |
| $F$ | Velocity profile | $N_B$ | Brownian motion parameter |
| $\rho$ | Density | $N_T$ | Thermo-phoresis parameter |

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
