# Peer review of "Buongiorno’s Nanofluid Model over a Curved Exponentially Stretching Surface"

_processes, doi:10.3390/pr7100665_

Round 1

Reviewer 1 Report

The abstract is potentially the most important part of your paper, abbreviations can not be understood by all readers such as bvp4c. I strongly suggest the authors rewrite the abstract to some extent involving the opening 2-3 sentences, 1-2 sentences should very briefly describe the method, 3-5 sentences on your results (optional) and The most important sentences are the final 2-3 sentences that are the conclusion of your study. 

From their introduction section, the authors attempted to modify  Buongiorno model in order to investigate the effects of Brownian motion and thermophoresis on the flow, heat and mass transfer over stretching channel.

; However the importance of their work has not been mentioned and sufficient explanation about their aims and objectives is not perceived in the introduction section. I recommend to just keep it simple mentioning the importance of the other people's work, and how your work can address some issues in this specific field. 

Figure 1 must be moved to a suitable place just after you first referred it in the main text.  

 Although reporting of any numerical results in the abstract is optional, the conclusion section has to have some direct demonstration of results.I suggest to revise this section as well, otherwise, their work would be questionable by potential readers. 

Author Response

Revised report

Q.1: The abstract is potentially the most important part of your paper, abbreviations can not be understood by all readers such as bvp4c. I strongly suggest the authors rewrite the abstract to some extent involving the opening 2-3 sentences, 1-2 sentences should very briefly describe the method, 3-5 sentences on your results (optional) and The most important sentences are the final 2-3 sentences that are the conclusion of your study.

Ans.: We improved the abstract section according to reviewer suggestions.

Q.2: From their introduction section, the authors attempted to modify  Buongiorno model in order to investigate the effects of Brownian motion and thermophoresis on the flow, heat and mass transfer over stretching channel.

Ans.: It is incorporated according to reviewer suggestions.

Q.3: However the importance of their work has not been mentioned and sufficient explanation about their aims and objectives is not perceived in the introduction section. I recommend to just keep it simple mentioning the importance of the other people's work, and how your work can address some issues in this specific field.

Ans.: We worked on the Buongiorno model to analyze the effects of of Brownian motion and thermophoresis parameters over curved channel theoretically and analytically, because this model is the theoretical models so we have not concentrated on practical point of view however experimental can you done our data. The thermal conductivity of the nanofluid improves the heat transfer rate and analyzes the effects of other physical assumptions.

Q.4: Figure 1 must be moved to a suitable place just after you first referred it in the main text.

Ans.: Improved according to the reviewer’s suggestion.

Q.5: Although reporting of any numerical results in the abstract is optional, the conclusion section has to have some direct demonstration of results.I suggest to revise this section as well, otherwise, their work would be questionable by potential readers.

Ans.: We improved the conclusion section and highlight the important results.

Reviewer 2 Report

This work used the steady flow model by Buongiorno and applied it to a permeable exponentially stretching, curved channel. The model's PDE system was transformed into an ODE (boundary value problem) using a similarity transformation and then solved numerically. They studied the effect of many parameters on the boundary layer thickness and concentration. They performed a very thorough investigation. 

I have made numerous comments, suggestions, and edits into the .pdf manuscript. In addition to those, I have a few additional comments/suggestions:

When introducing the mathematical model, please give explicit definitions for all parameters and an intuitive reason why studying each parameter is meaningful or useful.

Please try to formulate a metric for comparing different numerical solutions; e.g., can you answer "how different is each solution when lambda is changed from ___ to ___?" If this is too difficult to do so in a meaningful, concise way, please provide more insight into how subtle differences in solutions may be important for practical applications.

In the Conclusion/Discussion please provide more context about how your study compares to previous study, e.g., how close are your results to previous studies and what studies are those. How can your results be used in a practical way? 

Thanks!

Author Response

Revised report

This work used the steady flow model by Buongiorno and applied it to a permeable exponentially stretching, curved channel. The model's PDE system was transformed into an ODE (boundary value problem) using a similarity transformation and then solved numerically. They studied the effect of many parameters on the boundary layer thickness and concentration. They performed a very thorough investigation.

Q.1: I have made numerous comments, suggestions, and edits into the .pdf manuscript. In addition to those, I have a few additional comments/suggestions:

Ans.: We edit the comments which are highlighted in the pdf.

Q.2: When introducing the mathematical model, please give explicit definitions for all parameters and an intuitive reason why studying each parameter is meaningful or useful.

Ans.: we have introduced the mathematical model to analyze the Buongiorno model in order to investigate the effects of Brownian motion and thermophoresis on the flow, heat and mass transfer over stretching channel. Our work depends on the analytical and theoretical. The involved dimensionless parameters are meaningful in the field of industrial and engineering applications. Lots of applications have been seen in the literature.

Q.3: Please try to formulate a metric for comparing different numerical solutions; e.g., can you answer "how different is each solution when lambda is changed from ___ to ___?" If this is too difficult to do so in a meaningful, concise way, please provide more insight into how subtle differences in solutions may be important for practical applications.

Ans.: The physical parameter range is already given in our paper discussion sections. We find the solution of nonlinear coupled ordinary differential equation and find out the solution which are stable. We have presented the comparison of bvp4c, ND solve and shooting technique in Table 3. It is found that bvp4c method is best method as compared to ND solve and shooting technique. We have not worked out on the maximum range of the parameter, because each parameter has limitations physically as reported in the literature. So we have taken care of those parameters as suggested in the experiments. We have presented the theoretical idea which may be implemented experimentally. There are lots of applications in the fields of engineering and industrial.

Q.4: In the Conclusion/Discussion please provide more context about how your study compares to previous study, e.g., how close are your results to previous studies and what studies are those. How can your results be used in a practical way?

Ans.: We have used the Buongiorno model for the flow over a curved stretching surface. Thus the highlights and importance are presented.

Round 2

Reviewer 2 Report

Dear Authors,

Thank you for addressing our concerns.